# Association of single nucleotide polymorphisms with dyslipidemia in antiretroviral exposed HIV patients in a Ghanaian population: A case-control study

Christian Obirikorang[1]*, Emmanuel Acheampong[1,2], Lawrence Quaye[3], Joseph Yorke[4], Ernestine Kubi Amos-Abanyie[5], Priscilla Abena Akyaw[5], Enoch Odame Anto[1,2], Simon Bannison Bani[3], Evans Adu Asamoah[1], Emmanuella Nsenbah Batu[1]

1 Department of Molecular Medicine, School of Medical Science, Kwame Nkrumah University of Science and Technology (KNUST), Kumasi, Ghana, 2 School of Medical and Health Science, Edith Cowan University, Joondalup, Australia, 3 School of Allied Health Sciences, University of Development Studies, Tamale, Ghana, 4 Department of Surgery, Komfo Anokye Teaching Hospital, Kumasi, Ghana, 5 H3Africa Kidney Disease Research Project, Noguchi Memorial Institute for Medical Research, University of Ghana, Accra, Ghana

* emmanuelachea1990@yahoo.com

**Data Availability Statement:** All relevant data are within the manuscript and its Supporting Information files.

## Abstract

Dyslipidemia is a potential complication of long-term usage of antiretroviral therapy (ART) and also known to be associated with genetic factors. The host genetic variants associated with dyslipidemia in HIV patients on ART in Ghana have not been fully explored. The study constituted a total of 289 HIV-infected patients on stable ART for at least a year. Fasting blood was collected into EDTA tube for lipids measurement. Lipid profiles were used to define dyslipidemia based on the NCEP-ATP III criteria. HIV-infected subjects were categorized into two groups; those with dyslipidemia (cases) (n = 90; 31.1%) and without dyslipidemia (controls)(n = 199; 68.9%). Four candidate single nucleotide polymorphism (SNP) genes (ABCA1-rs2066714, LDLR-rs6511720, APOA5-rs662799 and DSCAML1-rs10892151) were determined. Genotyping was performed on isolated genomic DNA of study participants using PCR followed by a multiplex ligation detection reaction (LDR). The percentage of the population who had the rare homozygote alleles for rs6511720 (T/T), rs2066714 (G/G), rs10892151 (T/T) and rs662799 (G/G) among case subjects were 5.5%, 14.4%, 6.6% and 10.0% whiles 2.0% 9.1%, 6.5% and 4.0% were observed among control subjects. There were statistically significant differences in the genotypic prevalence of APOA5 (p = 0.0357) and LDLR polymorphisms (p = 0.0387) between case and control subjects. Compared to the AA genotype of the APOA5 polymorphisms, individuals with the rare homozygote genotype [aOR = 2.38, 95%CI(1.06–6.54), p = 0.004] were significantly associated with an increased likelihood of developing dyslipidemia after controlling for age, gender, treatment duration, CD4 counts and BMI. Moreover, individuals with the rare homozygous genotype of *ABCA1* (G/G) [aOR = 10.7(1.3–88.7), p = 0.0280] and *LDLR* (rs6511720) G>T [aOR = 61.2(7.6–493.4), p<0.0001) were more likely to have high levels of total cholesterol levels. Our data accentuate the presence of SNPs in four candidate genes

**Funding:** The authors received no specific funding for this work.

**Competing interests:** The authors have declared that no competing interests exist.

and their association with dyslipidemia among HIV patients exposed to ART in the Ghanaian population, especially variants in APOA5-rs662799 and LDLR rs6511720 respectively. These findings provide baseline information that necessitates a pre-symptomatic strategy for monitoring dyslipidemia in ART-treated HIV patients. There is a need for longitudinal studies to validate a comprehensive number of SNPs and their associations with dyslipidemia.

## Introduction

Global estimates report 37 million people living with Human Immunodeficiency Virus (HIV), out of which about 26 million reside in Sub-Saharan (SSA) [1]. In Ghana, HIV prevalence among adults aged 15–49 years has declined from about 2.4% in 2013 to 1.6% in 2015 according to the World Bank report [2]. The life expectancy of HIV-infected patients has increased remarkably due to the use of antiretroviral therapy (ART) as a standard of care [3–5]. Unfortunately, long term ART use is associated with a wide spectrum of metabolic disturbances such as lipodystrophy, insulin resistance, and dyslipidemia [6–8]. Dyslipidemia is defined by elevations in total cholesterol, low-density lipoprotein cholesterol (LDL-C), triglycerides and decreased high-density lipoprotein cholesterol (HDL-C).

The prevalence of dyslipidemia is reportedly higher in people living with HIV due to the effect of ART in Ghana [9–11]. The severity of dyslipidemia and the typical pattern of the lipid profile differ between and within the classes of antiretroviral (ARV) agents [12]. Lipid abnormalities have been reported to be frequently associated with HIV-infected individuals receiving protease inhibitors (PIs) and treatment-*naïve* HIV-infected patients, suggesting that HIV infection itself has a metabolic deleterious effect. Such reported side effects are not universal to all individuals on ART and may even vary in individuals with comparable ART, demographic, immunologic and virological characteristics [12–14]. This variability suggests that genetic factors and inherited predispositions may have a significant influence on the incidence of metabolic dysfunction [14, 15]. However, high HDL cholesterol and apolipoprotein A-I (*APOA-I*) have been directly associated with a better immunological outcome [16].

Low-density lipoprotein receptor (LDLR), positioned on chromosome 19p13.2 plays a significant role in lipoprotein metabolism by mediating the uptake of cholesterol through the binding and subsequent cellular uptake of apolipoprotein-E and B- constituting lipoproteins. Mutations have been detected in different domains of the LDLR which have a distinct effect on LDLR structure and function [17, 18]. ATP-binding cassette A1(ABCA1) plays a critical role in the reverse cholesterol transport system. Mutation in ABCA1, that encodes this protein, along with genes responsible for their transcription regulation, can lead to abnormality in the metabolism of lipids [19, 20]. Apolipoprotein A5 (APOA5) has been shown to be a key regulator of plasma triglycerides and there are several SNPs associated with the APOA5 gene [21, 22]. Moreover, HIV-infected patients who harbor polymorphisms of the DSCAML1 (Down syndrome cell adhesion molecule like-1) gene exhibit a less favorable lipids profile [23, 24]. Current studies have suggested the relationship between the level of lipids and LDLR, ABCA1 APOA5, and DSCAML1 polymorphisms [15, 25].

Nevertheless, the exact mechanism of dyslipidemia is not fully understood but is most likely multifactorial with the genetic variation being shown to account for about 43–83% of the variability of plasma lipoprotein levels in a normal healthy population [16, 26]. Therefore, from the genetic perspective, HAART-associated hyperlipidemia could be under the influence of

various forms of genetic polymorphisms, similar to that in non-HIV adults [27, 28]. Several single nucleotide polymorphisms (SNPs) that could account for a significant portion of the variation of blood lipoprotein concentrations have been identified through recent candidate gene studies and genome-wide association studies (GWAS)[15, 29].

The association between gene polymorphisms that may signal a predisposition to lipid abnormalities and clinical progression of HIV infection has not been thoroughly studied in Africa where the prevalence of HIV is on the increase. SNP prevalence differs by population and at present, the majority of the SNP-associated dyslipidemic studies among HIV patients have come from non-African countries with only a few of these studies emanating from Africa [15, 24, 30]. To the best of our knowledge, no published study has explored the genetic variants and markers associated with dyslipidemia in HIV-infected individuals on HAART in a Ghanaian population. This study, therefore, investigated the distribution of SNPs in four candidate genes that have had significant published lipid associations and their resultant associations with plasma lipid levels in Ghanaian HIV-infected patients on HAART. An understanding of the impact of host genetic factors on the prevalence of dyslipidemia in a cohort of HIV-infected individuals on HAART would promote interventions in the scaling up of treatment regimen.

## Material and methods

### Study design and subjects

This study comprised HIV-1 infected patients who were on ART regimen for at least one year, with either a protease inhibitor (PI) or non-nucleoside reverse-transcriptase inhibitor (NNRTI) backbone without any history of dyslipidemia, hypertension, and diabetes. The NCEP-ATP III criteria were used to defined dyslipidemia among the HIV seropositive [31]. The HIV seropositive subjects were categorized into two groups; those with dyslipidemia (cases) and without dyslipidemia (controls). The adjuvant antiretroviral drugs were stavudine, lamivudine, and zidovudine with priorities for inclusion being given to participants who consented to undergo biochemical and genetic testing. Pregnant women, patients being treated with lipid-lowering drugs and those with neurological conditions that prevented them from understanding the concept of the research were excluded from the study.

### Sample size determination

Based on previous report from a study conducted on the burden of dyslipidemia among adults in Ghanaian a population [32] and the lack of knowledge of the frequency of polymorphisms in the population, we assumed an expected proportion of 0.1 for exposure in HIV seropositive subjects with dyslipidemia, an assumed odds ratio of 2, a confidence interval of 95%, a power of identifying a significant difference between two groups, and 1:3 ratio, a total of 289 subjects were recruited.

### Data collection and biochemical analysis

A structured questionnaire was administered to each patient to obtain demographic information. Details on ARVs, time of diagnosis, duration on ARVs, CD4 counts were obtained from the medical folders of the patients. Fasting blood samples were collected for the analysis of lipid parameters and genomic DNA.

Blood samples were taken after an overnight (12–14 hours) fast into EDTA tubes for biochemical analysis. Fasting lipid panel including total cholesterol (TC), HDL-cholesterol and triglycerides (TG) were measured using Flexor junior (Vital Scientific, Dieren, Netherlands)

chemistry autoanalyzer. LDL-C was calculated from the Friedewald's formula, LDL-C = TC-(HDL-C- TG/2.2). Patient with triglycerides above 4.52 mmol/L was not included in the study since LDL-C was not directly measured and due to the deficit of the Friedewald's equation, which overestimates LDL-C levels when triglycerides are high. This is to reduce any bias that might affect the relationship found between LDL-C and dyslipidemia. The NCEP-ATP III criteria were used to define dyslipidemia as reduced HDL (<1.03 mmol/L in males; <1.29 mmol/L in females), raised TG ≥1.7 mmol/L, TC >6.2 mmol/L and LDL-C >3.37 mmol/L or specific treatment for such lipid abnormalities [31].

## Anthropometric and hemodynamic measures

Anthropometric measures such as weight and height were performed using an automated weighing scale. Portable height rod stadiometers were used for height measurements to the nearest centimeter. Body mass index (BMI) was defined as weight (kg)/height (m)$^2$. Blood pressure was measured using an automated sphygmomanometer (Omron M7 Intelli IT). Three consecutive readings of blood pressure measurements were taken from the patients' right arm and the mean of two closest values was recorded.

## Single nucleotide polymorphisms selection and genotyping

The four candidate SNPs (rs2066714, rs6511720, rs662799, and rs10892151) selected for this study have been shown in previous studies to be significantly associated with lipid serum abnormalities following a review of GWAS and PubMed reports of SNPs associated with dyslipidemia among HIV-infected individuals [24, 33–35]. Genomic DNA was isolated from EDTA-collected whole blood samples of study participants using the Qiagen midi kit prep as per manufacturer's protocol. Genotyping was carried out on isolated genomic DNA of study participants using a multiplex ligation detection reaction (LDR), a sequence-specific genotyping method that has been used efficiently in polymorphism typing and detection of mutations in disease genes [36]. Triplex multiplex reaction assays were setup with products being run on 10% polyacrylamide gel for LDR genotype observation (58 to 90 base pairs). The LDR products differed in sizes of about 8 base pairs for each triplex (Fig 1). The runs were controlled with Amelogenin XY (AME XY) to determine a successful reaction and gender-confirming for study participants.

Allele specific LDR products are assigned by color and size following 10% polyacrylamide gel electrophoresis. Each band as shown in the image represents a specific genotype of each allele. At the bottom of the gel are leftover LDR probes from the reaction mix as represented by shared common bands (Fig 1).

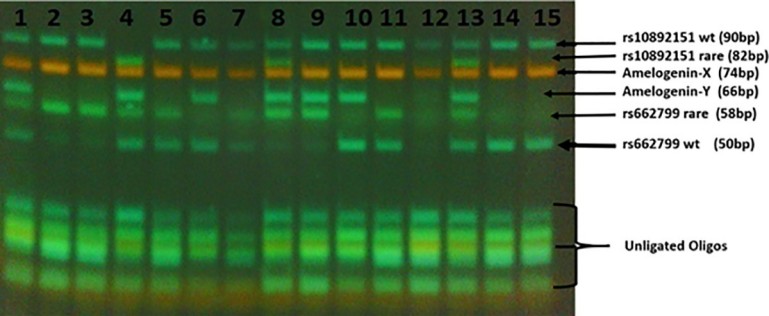

**Fig 1. A representative gel depicting a genotype assay using triplex ligation detection reaction.**

## Ethical consideration

The study protocol was approved by the Committee of Human Publication and Research Ethics of the School of Medical Sciences, Kwame Nkrumah University of Science and Technology, Kumasi, Ghana. All participants gave written informed consent and were assured that the information gathered was to be used strictly for research and academic purpose only. In addition, respondents were given the freedom to opt-out at any time they thought they could not continue with the study.

## Data management and statistical analysis

The NCEP-ATP III criteria were used to defined dyslipidemia among the HIV seropositive [32]. The HIV seropositive subjects were categorized into two groups; those with dyslipidemia (cases) and without dyslipidemia (control). Microsoft Excel software was used to set up a database, and to avoid entry error, the double-entry method was used, and data were analyzed using SPSS version 25 and R program where appropriately. Parametric continuous were analyzed with a t-test and expressed as mean ± standard deviation (SD) after checking for normality with the Kolmogorov-Smirnov test.

ANOVA was used to compare continuous variables from more than two groups with Yates post-test, the chi-square test was to compare differences between more than three groups for categorical variables while Fisher exact was used to assess differences between two groups for categorical variables Allele frequencies were estimated by gene counting. Deviations in the genotype frequencies from the Hardy-Weinberg equilibrium (HWE) were tested using the chi-square ($\chi^2$) analysis. Initial univariate binary logistic models were performed to determine the association between SNPs and dyslipidemia, followed by adjusted multivariate binary logistic models controlling for age, gender, BMI, CD4 counts and duration of HIV infection. Adjusted multivariate binary logistic models were used also to identify SNPs independently associated with lipid abnormalities holding confounding variables constant. A p-value of less than 0.05 was considered statistically significant.

## Results

**Table 1** shows a comparison of demographic, hemodynamic indices, lipid parameters, dyslipidemic indices between cases and control subjects. Dyslipidemia was found in 31.1% (90/289). A statistically significant difference between case and control subjects was observed with regards to age (p = 0.0328). There were more females than males in the study population (p>0.05). The average duration of treatment among the subjects was 4 years. There were statistically significant differences between the case and control subjects in relation to CD4 counts (p = 0.0010) and BMI (p<0.0001). The case subjects had significantly higher levels of TC (p<0.000) LDL-C (p<0.0001) and CR (p<0.0001) compared to the control subjects. Compared to HIV seropositive subjects with dyslipidemia, those without dyslipidemia had significantly increased levels of HDL-C (0.7 ±0.3 vs. 1.2±0.5, p<0.0001).

**Table 2** shows the frequency distribution of the genotypes and alleles of the four SNPs. The percentage of the population who had the rare homozygote alleles for rs6511720 (T/T), rs2066714 (G/G), and rs10892151 (T/T) and rs662799 (G/G) among the case subjects were 5 .5%, 14.4%, 6.6% and 10.0% whiles 2.0% 9.1%, 6.5% and 4.0% were observed among the control subjects. Statistically significant differences in allelic frequency were observed for DSCAML1 (p = 0.0008)and APOA5 (p = 0.0251) among case subjects, and ABCA1 (p = 0.0010) and DSCAML1 (p = 0.0084) among control subjects. Moreover, chi-square analysis reveals significant differences in the genetic frequencies of LDLR (p = 0.0387) and APOA5 (p = 0.0353) polymorphisms respectively.

**Table 1. Comparison of the general characteristics of study participants.**

| Variables | Cases (n = 90) | Controls (n = 199) | P-value |
|---|---|---|---|
| Age (years) | 40.9±10.8 | 39.2±2.0 | 0.0328 |
| Gender [b] | | | 0.7535 |
| Female | 73(81.1%) | 157(78.9%) | |
| Male | 17(18.9%) | 42(21.1%) | |
| Duration of treatment | 4.3±2.6 | 4.2±2.9 | 0.7796 |
| CD4 counts | 420.7±28.9 | 410.6±21.2 | 0.0010 |
| **Blood pressure (mmHg)** | | | |
| Systolic | 112.5±13.8 | 114.8±17.6 | 0.2739 |
| Diastolic | 79.9±7.6 | 81.5±8.6 | 0.2309 |
| Anthropometric indices | | | |
| Weight (kg) | 76.9±13.8 | 76.6±14.5 | 0.8688 |
| Height (m) | 1.58±0.18 | 1.60±0.13 | 0.2861 |
| Body mass index (BMI) (kg/m$^2$) | 33.8±2.4 | 31.5±1.1 | <0.0001 |
| **Lipid parameters (mmol/L)** | | | |
| Total Cholesterol | 5.2±0.7 | 4.0±0.8 | <0.0001 |
| Triglycerides | 1.4±0.7 | 1.4±0.6 | 0.9999 |
| HDL-Cholesterol | 0.7±0.3 | 1.2±0.5 | <0.0001 |
| LDL-Cholesterol | 4.1±0.6 | 2.5±0.8 | <0.0001 |
| VLDL-Cholesterol | 0.6±0.34 | 0.6±0.35 | 0.9999 |
| Coronary Risk | 10.5±0.3 | 6.2±4.8 | <0.0001 |
| **Dyslipidemic indices** | | | |
| High triglyceride [b] | 8(8.9%) | 26(13.1%) | 0.4986 |
| High total cholesterol [b] | 10(11.1%) | 4(2.0%) | 0.0184 |
| Low HDL-Cholesterol [b] | 90(100.0%) | 107(53.8%) | <0.0001 |
| High LDL-Cholesterol [b] | 90(100.0%) | 14(7.0%) | <0.0001 |

Cases: HIV seropositive with dyslipidemia, control: HIV seropositive without dyslipidemia

[b]Fisher exact test, Dyslipidemia is defined as the presence of at least one NCEP-ATP III criteria reduced HDL (<1.03 mmol/L in males; <1.29 mmol/L in females), raised TG ≥1.7 mmol/L, TC >6.2 mmol/L and LDL-C >3.37 mmol/L p<0.05 is considered statistically significant.

### Total Cholesterol (TC)

The case subjects who had the rare homozygous genotype for LDLR (p<0.0001), ABAC1 (p = 0.0287) DSCAML1 (p = 0.0003) and APOA5 (p = 0.0151) had a significantly higher level of TC compared to the combined heterozygous and non-carriers' genotypes [Table 3].

### Triglycerides (TG)

Statistically significant differences were observed in TG levels for DSCAML1(p = 0.0006) and LDLR (p = 0.0178) polymorphisms between homozygous rare genotype and heterozygous and non-carriers' genotypes among the case subjects. Similar patterns were observed for DSCAML1 polymorphisms (p = 0.0133) among the control subjects [Table 3].

### HDL Cholesterol (HDL-C)

There were significant differences in HDL-C levels with regards to APOA5 polymorphisms (p = 0.0075) in case subjects. Thus, the combined heterozygous and non-carriers genotype had high levels of HDL-C compared to the homozygous genotypes [Table 3].

**Table 2. Genotypic and allelic frequencies of polymorphisms of the Studied Population.**

| Polymorphisms | Cases | | | Controls | | | |
|---|---|---|---|---|---|---|---|
| | n (%) | Allelic frequency | | n (%) | Allelic frequency | | #P-value |
| LDLR (rs6511720) G>T | | G | T | | G | T | **0.0387** |
| | | 0.81 | 0.19 | | 0.89 | 0.11 | |
| G/G | 61(67.8) | | | 159(79.9) | | | |
| G/T | 24(26.7) | | | 34(18.1) | | | |
| T/T | 5(5.5) | | | 4(2.0) | | | |
| HWE-P¥ | | 0.28184 | | | 0.1875 | | |
| ABCA1 (rs2066714) A>G | | G | A | | G | A | 0.3876 |
| | | 0.73 | 0.27 | | 0.78 | 0.22 | |
| G/G | 13(14.4) | | | 18(9.1) | | | |
| A/G | 23(25.6) | | | 53(26.6) | | | |
| A/A | 54(60.0) | | | 128(64.3) | | | |
| HWE-P¥ | | 0.0008 | | | 0.0010 | | |
| DSCAML1 (rs10892151) C>T | | C | T | | C | T | 0.9585 |
| | | 0.8 | 0.2 | | 0.81 | 0.19 | |
| C/C | 60(66.7) | | | 136(68.3) | | | |
| C/T | 24(26.7) | | | 50(25.2) | | | |
| T/T | 6(6.6) | | | 13(6.5) | | | |
| HWE-P¥ | | 0.1138 | | | 0.0084 | | |
| APOA5 (rs662799) A>G | | A | G | | A | G | 0.0353 |
| | | 0.76 | 0.24 | | 0.84 | 0.16 | |
| A/A | 56(62.2) | | | 142(71.4) | | | |
| A/G | 25(27.8) | | | 51(25.6) | | | |
| G/G | 9(10.0) | | | 6(3.0) | | | |
| HWE-P¥ | | 0.0251 | | | 0.1625 | | |

Cases: HIV seropositive with dyslipidemia, control: HIV seropositive without dyslipidemia, HWE-P: Hardy-Weinberg equation p-value

#P-value represents chi-square test to compare genotype frequency between cases

¥P-values for Chi-squared test for variant allelic frequency based on HWE; If p<0.05 means it not consistent with HWE

## LDL Cholesterol (LDL-C)

There were significant increased LDL-C levels in the rare-allelic subjects for LDLR (p<0.0001) and DSCAML1 (p = 0.0078) polymorphisms compared to participants with heterozygous and homogenous genotypes together among the case subjects [Table 3].

Individuals with G/G genotype for APOA5 (rs662799) polymorphisms were significantly associated with an increased likelihood of developing dyslipidemia upon both univariate [OR = 3.54, 95% CI(1.27–10.25; p = 0.0208)] and multivariate [OR = 2.38, 95% CI(1.06–6.54; p = 0.0093)] logistic regression analyses holding other variables constant. A similar trend was observed for rare homozygous genotype for LDLR polymorphisms; however, no statistically significant association was noted [Table 4].

Table 5 shows the association between SNPs with lipid abnormalities after controlling for age, gender, duration of treatment, CD4 counts and BMI respectively. Individuals with the rare homozygous genotype of *ABCA1* (G/G) [aOR = 10.7(1.3–88.7), p = 0.0280] and *LDLR* (rs6511720) G>T [aOR = 61.2(7.6–493.4), p<0.0001) were more likely to have high levels of TC levels. Moreover, subjects with T/T genotype of *APOA5* (G/G) polymorphism were associated with increased levels of LDL-C [aOR = 2.2(1.4–6.0), p = 0.014].

**Table 3. Comparison of lipid parameters among study participants based on polymorphisms.**

| Variables | Cases | | | | Controls | | | |
|---|---|---|---|---|---|---|---|---|
| **Polymorphisms** | **TC** | **TG** | **HDL-C** | **LDL-C** | **TC** | **TG** | **HDL-C** | **LDL-C** |
| | **(mmol/L)** | **(mmol/L)** | **(mmol/L)** | **(mmol/L)** | **(mmol/L)** | **(mmol/L)** | **(mmol/L)** | **(mmol/L)** |
| LDLR (rs6511720) | | | | | | | | |
| G>T | | | | | | | | |
| GG/GT | 5.0±0.6 | 1.4±0.6 | 0.7±0.3 | 4.0±0.5 | 3.9±0.8 | 1.4±0.8 | 1.4±0.5 | 2.4±0.8 |
| T/T | 6.6±1.0 | 2.1±1.8 | 0.6±0.4 | 5.5±0.6 | 4.5±1.4 | 1.6±0.6 | 1.1±0.5 | 2.7±1.2 |
| p-value | **<0.0001** | **0.0178** | 0.895 | **<0.0001** | 0.1832 | 0.7018 | 0.2125 | 0.4866 |
| ABCA1 (rs2066714) | | | | | | | | |
| A>G | | | | | | | | |
| AA/AG | 5.1±0.7 | 1.3±0.7 | 0.6±0.3 | 4.1±0.1 | 3.8±0.9 | 1.4±0.9 | 1.2±0.6 | 2.4±0.8 |
| G/G | 5.5±0.8 | 1.8±0.7 | 0.8±0.4 | 4.4±0.6 | 4.2±1.1 | 1.4±0.06 | 1.1±0.4 | 2.7±0.9 |
| p-value | **0.0287** | 0.1148 | 0.2213 | 0.0968 | 0.247 | 0.8847 | 0.4001 | 0.3085 |
| DSCAML1 (rs10892151) | | | | | | | | |
| C>T | | | | | | | | |
| CC/CT | 4.8±0.5 | 1.2±0.4 | 0.6±0.5 | 4.1±0.6 | 4.0±0.9 | 1.4±0.6 | 1.2±0.5 | 2.3±0.6 |
| T/T | 6.1±1.2 | 2.3±1.5 | 0.9±0.4 | 4.8±0.9 | 4.3±1.0 | 1.9±1.1 | 1.0±0.6 | 2.9±0.6 |
| p-value | **0.0003** | **0.0006** | 0.1036 | **0.0078** | 0.1946 | **0.0133** | 0.3933 | 0.0665 |
| APOA5 (rs662799) | | | | | | | | |
| A>G | | | | | | | | |
| AA/AG | 5.0±0.2 | 1.4±0.7 | 0.6±0.3 | 4.2±0.6 | 4.1±0.7 | 1.5±0.7 | 1.2±0.4 | 2.5±0.8 |
| G/G | 5.6±0.9 | 1.3±0.6 | 0.9±0.2 | 4.5±0.8 | 4.4±1.2 | 1.7±0.9 | 1.0±0.6 | 2.8±0.9 |
| p-value | **0.0151** | 0.7388 | **0.0075** | 0.1017 | 0.1563 | 0.3039 | 0.5685 | 0.2911 |

TC: Total cholesterol, TG: Triglyceride, HDL-C: High density lipoprotein cholesterol, LDL-C: Low density lipoprotein cholesterol, Cases: HIV seropositive with dyslipidemia, Control: HIV seropositive without dyslipidemia, Data is presented as mean±standard deviation, T-test was performed to obtained p-values, p<0.05 significant

**Table 4. Association between single nucleotide polymorphisms and the prevalence of dyslipidemia.**

| Variables | Dyslipidemia | | | |
|---|---|---|---|---|
| | **Univariate logistic model** | | **Multivariate logistic model** | |
| **Polymorphisms** | **OR (95% CI)** | **P-value** | **aOR (95% CI)** | **P-value** |
| LDLR (rs6511720) | | | | |
| GG/GT | 1 | | 1 | |
| T/T | 2.87(0.83–9.50) | 0.1427 | 2.23(0.61–8.18) | 0.2260 |
| ABCA1 (rs2066714) | | | | |
| GG/AG | 1 | | 1 | |
| A/A | 1.70(0.81 3.57) | 0.2171 | 1.56(0.68–3.57) | 0.2910 |
| DSCAML1 (rs10892151) | | | | |
| CC/CT | 1 | | 1 | |
| T/T | 1.02(0.38–2.63) | 0.8944 | 0.68(0.22–2.11) | 0.5090 |
| APOA5 (rs662799) | | | | |
| AA/AG | 1 | | 1 | |
| G/G | 3.54(1.27–10.25) | 0.0208 | 2.38((1.06–6.54 | 0.0093 |

Cases: HIV seropositive with dyslipidemia, Control: HIV seropositive without dyslipidemia, Odds ratio was calculated for each single nucleotide polymorphism using rs6511720-GG, rs2066714-GG, rs10892151-CC, and rs662799-AA genotypes as a referent genotype OR = Odds ratio, CI = Confidence interval. A multivariate logistic model was performed controlling for age, gender, duration of treatment, CD4 counts and BMI, p<0.05 considered statistically significant

**Table 5. Association of SNPs with individual lipid abnormalities.**

| Variables | High TC (mmol/L) | High TG (mmol/L) | Low HDL-C (mmol/L) | High-LDL-C (mmol/L |
|---|---|---|---|---|
| **Polymorphisms** | **aOR (95% CI)** | **aOR (95% CI)** | **aOR (95% CI)** | **aOR (95% CI)** |
| LDLR (rs6511720) T/T | 61.2 (7.6–493.4)[†] [††] | 0.9 (0.1–1.0) | 0.9 (0.2–3.9) | 2.7 (0.7–10.6) |
| ABCA1 (rs2066714) G/G | 10.7 (1.3–88.7) [†] | 1.8 (0.6–5.3) | 1.3 (0.5–3.2) | 1.7 (0.8–3.9) |
| DSCAML1 (rs10892151) T/T | 2.3 (0.3–17.2) | 1.7 (0.5–6.2) | 0.8 (0.3–4.8) | 0.9 (0.3–2.8) |
| APOA5 (rs662799) G/G | 5.8 (0.8–44.9) | 0.9 (0.2–4.6) | 1.4 (0.4–4.8) | 2.2 (1.4–6.0) [†] |

TC = Total cholesterol, TG = Triglycerides, HDL-C = High Density Lipoprotein Cholesterol, LDL-C = Low Density Lipoprotein Cholesterol, Odds ratio was calculated for the rare recessive polymorphism of rs6511720, rs2066714, rs10892151 and rs662799 genotypes using a combination of carriers and homozygotes as a referent genotype, aOR = Adjusted odds ratio, CI = Confidence interval. A multivariate logistic model was performed controlling for age, gender, duration of treatment, CD4 counts and BMI

[†] $p < 0.05$

[†] [††] $p < 0.0001$, $p < 0.05$ = statistically significant

**S1 Table** shows the demographic, clinical and metabolic characteristics of participants stratified on the type of ART (NNRTI vs. PI). There were 3.5% (10/289) on PI-based treatment whiles 279 (96.4%) were on NNRTI-based treatment. No statistically significant association was observed between NNRTI AND PI- based treatment in relation to age (p = 0.2569), gender (p = 0.6929), duration of treatment (p = 0.2092), systolic (p = 0.7638) and diastolic pressure (p = 0.0865), weight (p = 0.2122), height (p = 0.7221) and BMI (0.6825) respectively. Seventy-five (15.6%) patients had hypertriglyceridemia, 24.9% had hypercholesterolemia and 71.9% had low HDL-C in relation to metabolic parameters. PI-based subjects had significantly higher levels of total cholesterol (6.09±0.56 vs. 4.34±0.96, p = 0.0001) and LDL cholesterol (4.60±0.55 vs. 2.95±1.16, p = 0.0001) and triglycerides (2.61±0.65 vs. 1.43±0.69, p = 0.0001) compared to NNRTI-based subjects.

## Discussion

Long term usage of ART has been implicated in several metabolic effects including dyslipidemia. However, this side effect varies among individuals on ART with comparable clinical and demographic characteristics. As such, inherited predispositions and genetic factors have been implicated in the cause of this metabolic alteration. The present study, therefore, assessed the prevalence of SNPs and their associations with dyslipidemia in HIV patients on ART in Ghana. Four candidate SNPs (rs2066714, rs6511720, rs662799, and rs10892151) reviewed from PubMed were analyzed for associations with dyslipidemia among HIV-infected individuals.

Several studies have shown the association between dyslipidemia and HIV patients on ART [37, 38]. The prevalence rate of dyslipidemia among HIV patients on ART in this current study was 31.1% with low HDL-C and high TC levels being the commonest lipid abnormalities. The observed prevalence of dyslipidemia is comparable to the range of reports from previous studies by Obirikorang et al.[9] and Ngala et al. [10] in Ghana and Kodogo et al. [10] in Zimbabwe. However, it is lower compared to 78.9% prevalence rate reported by Limas et al. [39] in a cross-sectional study among Brazilian HIV individuals on ART. Furthermore, our observed results for low HDL-C, high TC and LDL-C levels among HIV patients on ART are consistent with other studies [40, 41]. Results from the present study revealed that subjects who received PI-based treatment had significantly increased levels of TC, LDL cholesterol and triglycerides levels compared to the non-PI-based subjects. These findings are consistent with

several case reports [42, 43] and cross-sectional studies [44, 45] which have reported that PI exposures are associated with hypercholesterolemia and hypertriglyceridemia.

Our study demonstrated that the presence of the homozygous recessive/mutant genes for *LDLR*, *ABCA1*, *DSCAML1* and *APOA5* genotypes among our study participants were very low [Table 2]. Similar low frequencies in APOA5 and LDLR homozygotes mutant genes were reported by Lazzaretti et al. [15] in a cross-sectional study among HIV-infected patients on ART in the Brazilians population. Genome-wide associations studies have identified LDL-R SNP rs6511720 (G>T), which is located in intron-1 of the gene, to be associated with lower plasma levels of LDL-C and a lower risk of CHD [35]. Data from the GLGC consortium sug-gested that *LDLR* rs6511720 minor allele is prevalent in about 10% of the population and have established that the allele is protective, being associated with lower levels of LDL-C [46].

Notwithstanding that, the frequency of recessive genes was low in this study population, these findings are consistent with previous reports by Wang et al., [19] and Aragones et al. [24] who did not find any significant differences in their studies with respect to prevalence of the DSCAML1 and ABCA1 genotypes. Our findings demonstrated that the allelic frequencies for ABCA1 and APOA5 polymorphisms were in not equilibrium with the Hardy-Weinberg equa-tion among the case subjects as well as ABCA1 and DSCAML1 polymorphisms among the control subjects. The low frequencies of the homozygotes of the rare alleles of these SNPS may have contributed to the observed deviations.

A longitudinal study conducted by Rotgar et al. [47] validated the contribution of 42 SNPs to dyslipidemia among the HIV-infected population treated with ART. The authors reported that the degree of the contribution of SNPs and ART to dyslipidemia are similar and therefore genetic information should be considered in addition to the dyslipidemic effect of ART agents [47]. Results from the present study revealed that individuals with rs6511720 T/T genotype recorded increased levels of TC, TG, and LDL-C compared to the combination of non-carriers and heterozygous genotypes (GG/GT) [Table 3]. A further multivariate logistic model analysis showed a strong association of LDLR polymorphisms with high levels of TC holding all other confounding variables constants. A study by Lazzaretti et al. [15] reported that *LDLR* intron 19G>T (rs6511720) did not contribute to the plasma lipid levels in their dataset, and hence may reflect a limited effect of this SNPs in HIV-infected patients. The inconsistency could be due to different geographical settings and thereby calls for more research in these SNPs among African descendants.

In this study, ABCA1 (rs2066714) G/G genotype was associated with an increased probabil-ity of having dyslipidemia. Furthermore, subjects with the homozygotes of the rare alleles of ABCA1 have increased levels of TC compared to the common allelic (AA/AG) genotypes. As a further matter, subjects carrying the minor allelic variant of ABCA1 were more likely to have low levels of HDL cholesterol among the HIV population in the present study though no statis-tical significance was observed. These observations are congruent with previous reports dem-onstrated in literature that ABCA1 is associated with familial HDL deficiency. We should mention here that low HDL-C was one of the commonest lipid abnormalities observed in this study. Moreover, ABCA1 mediates the efflux of cellular cholesterol and phospholipids unto ApoA-I and thereby plays a central role in regulating cellular cholesterol homeostasis, and forming HDL[48, 49].

Individuals with T/T genotype for the *DSCAML1* (rs10892151) polymorphisms among the cases in this study had increased levels of TC, TG, and LDL-c compared with the other com-bined genotypes. Further analysis indicated that individuals with the homozygous minor recessive gene (carriers) (T/T) were more likely to have increased levels of TC and LDL-C. These observed findings are in parallel with previous reports by Aragones et al. [24] who reported a strong association between the expression of the rs10892151 T allelic variant and

dyslipidemia, mostly hypertriglyceridemia and decreased HDL-cholesterol levels. In contrast, Pollin et al., [23] observed that rs10892151 T carriers had lower fasting and postprandial serum triglycerides values than non-carriers, and they found a linkage disequilibrium with an APOC3 null mutation, which was likely the result of a founder effect in their high-fat feeding intervention study. Evidence provided in the literature shows that carriers of this null mutation have low circulating apolipoprotein (Apo) C-III levels and reduced fasting and post-prandial triglyceride concentrations [23], which is likely due to the well-established function of Apo C-III as an inhibitor of lipoprotein lipase[50].

This current study showed that carriers of the minor allelic variant of the *APOA5* (rs662799) A>G SNP gene, were 2 times more likely to developed dyslipidemia. Moreover, individuals with homozygotes (G/G) have significantly increased levels of TC and LDL cholesterol compared to wild type and heterozygote combined (AA/AG). Previous studies have found rs662799 to be associated with elevated plasma triglyceride levels as well as HDL-C and total cholesterol [21]. Two polymorphisms in the *APOA5* gene, −1131T>C and S19W (56C>G), have already been shown to be associated with elevated triglyceride levels in different populations[51, 52]. This study, however, considered the *APOA5* (rs662799) A>G variant and found that HIV individuals with at least one G allele had higher TC and LDL cholesterol levels.

Lazzaretti et al. [15] reported that *APOA5* −1131T>C (rs662799) was associated with plasma triglycerides (TG) and low-density lipoprotein cholesterol levels (LDL-C) as well as high-density-lipoprotein cholesterol levels. To the best of our knowledge, this study is the first to investigate the *APOA5* (rs662799) A>G variant in HIV patients in a Ghanaian population. Another study by Echeverria et al. [53] reported that polymorphisms in genes associated with the development of atherogenic dyslipidemia, especially variants in the *APOA5* gene, can influence the circulating CD4 T-cell levels in chronically HIV-infected patients. Although a previous study has reported the effect of APOA5 on CD4 levels, individuals with the minor allelic variant of *APOA5* (rs662799) A>G have a significant likelihood of developing dyslipidemia in HIV subjects on ART after controlling for CD4 counts as well as age, gender, duration of treatment and BMI.

This study has strength in it possibly being the first study to assess the prevalence of SNPs and their associations with dyslipidemia in the Ghanaian population as a proof of concept. Although previous reports have shown an uneven continuous rate of the incidence of HIV in the Ghanaian population with more than 60% of people living with HIV are females [54], the proportion of HIV- infected females in this study population was much higher than males which could affect the generalizations of our findings to the HIV population in Ghana. In addition, our study is limit by the fact that the number of investigated polymorphisms is not comprehensive, notwithstanding, this is a baseline study for future exploratory analysis of SNPs and their associations with dyslipidemia among HIV patients on ART in Ghana.

## Conclusion

This study has highlighted the evidence that SNPs in four candidate genes are present in HIV patients exposed to ART in the Ghanaian population. Dyslipidemia remains prevalent among HIV patients. SNPs were found to associate with dyslipidemia especially variants in APOA5-rs662799 and LDLR-rs651172 respectively. This finding provides baseline information that necessitates a pre-symptomatic strategy for monitoring dyslipidemia in ART-treated HIV patients Findings from this study should be validated in a longitudinal case-control study considering the disease and its therapeutic implications. These candidate SNPs if validated would help in serving as potential biomarkers to detect individuals at risk for dyslipidemia.

## Supporting information

**S1 Dataset. SPSS sheet of a dataset on which conclusions of this manuscript were made.**
(SAV)

**S1 Table. Demographic, clinical and metabolic characteristics of participants stratified on the type of ART (NNRTI vs. PI).**
(DOCX)

## Acknowledgments

The authors thank Prof. David T. Burke and Jodi Wilkowski of the Department of Human Genetics, University of Michigan Medical School, Michigan, the USA for immense help. We also wish to express our profound gratitude to all HIV patients who actively participated in this study.

## Author Contributions

**Conceptualization:** Christian Obirikorang, Emmanuel Acheampong, Priscilla Abena Akyaw, Enoch Odame Anto, Evans Adu Asamoah, Emmanuella Nsenbah Batu.

**Data curation:** Christian Obirikorang, Priscilla Abena Akyaw, Enoch Odame Anto, Evans Adu Asamoah, Emmanuella Nsenbah Batu.

**Formal analysis:** Emmanuel Acheampong, Lawrence Quaye, Enoch Odame Anto, Emmanuella Nsenbah Batu.

**Methodology:** Christian Obirikorang, Emmanuel Acheampong, Ernestine Kubi Amos-Abanyie, Evans Adu Asamoah.

**Project administration:** Emmanuel Acheampong.

**Resources:** Ernestine Kubi Amos-Abanyie.

**Supervision:** Christian Obirikorang, Emmanuel Acheampong, Joseph Yorke, Simon Bannison Bani.

**Validation:** Christian Obirikorang, Emmanuel Acheampong.

**Writing – original draft:** Christian Obirikorang, Emmanuel Acheampong, Lawrence Quaye, Joseph Yorke, Ernestine Kubi Amos-Abanyie, Priscilla Abena Akyaw, Enoch Odame Anto, Simon Bannison Bani, Evans Adu Asamoah, Emmanuella Nsenbah Batu.

**Writing – review & editing:** Christian Obirikorang, Emmanuel Acheampong, Lawrence Quaye, Joseph Yorke, Ernestine Kubi Amos-Abanyie, Enoch Odame Anto, Simon Bannison Bani, Evans Adu Asamoah, Emmanuella Nsenbah Batu.

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
