## [Decision Letter · Decision Letter 0]

18 Sep 2019

PONE-D-19-22071

Association of Single Nucleotide Polymorphisms with Dyslipidemia in Antiretroviral Exposed HIV Patients: A Case-Control Study in a Ghanaian population

PLOS ONE

Dear Mr Acheampong,

Thank you for submitting your manuscript to PLOS ONE. After careful consideration, we feel that it has merit but does not fully meet PLOS ONE’s publication criteria as it currently stands. Therefore, we invite you to submit a revised version of the manuscript that addresses the points raised during the review process. We would appreciate receiving your revised manuscript by Nov 02 2019 11:59PM. To enhance the reproducibility of your results, we recommend that if applicable you deposit your laboratory protocols in protocols.io, where a protocol can be assigned its own identifier (DOI) such that it can be cited independently in the future. For instructions see: http://journals.plos.org/plosone/s/submission-guidelines#loc-laboratory-protocols

We look forward to receiving your revised manuscript.

Kind regards,

Yong-Gang Yao

Academic Editor

PLOS ONE

Reviewers' comments:

Reviewer's Responses to Questions

**Comments to the Author**

1. Is the manuscript technically sound, and do the data support the conclusions?

Reviewer #1: Partly

Reviewer #2: Partly

2. Has the statistical analysis been performed appropriately and rigorously? 

Reviewer #1: Yes

Reviewer #2: No

3. Have the authors made all data underlying the findings in their manuscript fully available?

Reviewer #1: No

Reviewer #2: No

4. Is the manuscript presented in an intelligible fashion and written in standard English?

Reviewer #1: No

Reviewer #2: No

5. Review Comments to the Author

Reviewer #1: Though obviously suppressing HIV transmission and extending the life expectancy of HIV-infected patients, antiretroviral therapy (ART) also shows some side effects on human health. For example, long-term use of ART is associated with metabolic disturbances, such as dyslipidemia. In this study, the author studied the relationship of gene variants with dyslipidemia in HIV-infected patients receiving ART. Their data suggested that protease inhibitor treatment might be associated with higher levels of total cholesterol (TC), low-density lipoprotein cholesterol (LDL-C), and triglycerides [TG], compared with other ART treatment. In addition, four SNPs from four genes (ABCA1, LDLR, APOA5, and DSCAML1) are associated with dyslipidemia in HIV-infected patients receiving ART.

Major issues:

1. In the introduction, to easily make the reader understand the whole paper, the author had better explain why they selected these four SNPs to be studied.

2. The author said “Significantly increased levels of total cholesterol (TC) low-density lipoprotein cholesterol (LDL-C) and triglycerides [TG] were observed in protease inhibitor-based (PI) treated case subjects compared to non-PI-based case subjects” in the abstract. However, the similar presentation is deficient in the result section. Please add the corresponding content in the result section.

3. To make the reader better understand the relationship of different ART with dyslipidemia in HIV-infected patients, the author may as well describe the number of HIV-infected patients who receive PI or NNRTI.

Minor issues:

1. In the first paragraph of the result section, “All but HDL-cholesterol levels were significant levels in the control subjects compared to case subjects [ 1.44±0.33 vs. 0.99±0.53, p<0.0001].” should be “All but HDL-cholesterol levels were significantly lower levels in the control subjects compared to case subjects [ 1.44±0.33 vs. 0.99±0.53, p<0.0001].”

2. In Table 5, the author should clarify the definition of PRR in the sentence “DSCAML1 (T/T) [PRR=11.46(3.65-43.56), p<0.0001)”.

Reviewer #2: In the present study, Acheampong et al analyze the potential association of genetic variation in four different genes with the existence of dyslipidemia in HIV-infected patients on stable cART from Ghana.

Overall Major concern

The main concern of the study is that the design is very confusing. The authors state that the study is a case-control study. However there seems to be some confusion in the groups of selected individuals and their assignment to either cases or controls. In a case-control study, cases are those individual presenting the disease (in this case dyslipidemia) and controls are those not presenting the disease. The objective of the study (as stated by the authors) is to “..investigate the distribution of SNPs in four candidate genes and resultant association with plasma lipid levels in Ghanian HIV-infected patients on HAART”. For this purpose cases should be HIV patients with dyislipidemia and controls HIV patients without dyslipidemia. However, the authors recruit a population of HIV-seronegative adults without dyislipidemia as controls (in the author’s own words: “…alongside age-matched control subjects with no history of HIV, dyslipidemia, hypertension, and diabetes..”). In my opinion this is not the ideal control population as I have explained above. The inclusion of a population of individuals seronegative for HIV (with and without dyslipidemia) would serve to answer the question: Are the SNPs associated with dyslipidemia in the HIV population the same as those in the non-HIV population? However this is not the question that the objective addresses as the authors state (see above). Moreover, thereafter in the results section, the authors report that 18,3% of controls (ie. HIV-seronegative individuals) presented dyslipidemia. Thus there is a contradiction between the results and the inclusion criteria for the control population (“age-matched control subjects with no history of HIV, dyslipidemia, hypertension, and diabetes”). Also, the only data given for the control population is the prevalence of dyslipidemia and the distribution of allelic and genotypes frequencies for the different SNPs, but no data are given about the association of SNPs and dyslipidemia in this control population. Lastly, there is no discussion at all about the findings in the HIV-seronegative (“control”) population. Thus what is the reason to include this “control” population?.

In summary the authors must clarify this and explain the reason to include the “control” (HIV-seronegative individuals) population.

Specific concerns

1.- Introduction:

.- Some references are not the most adequate, for example references #3-5, reference #1 (this reference is very old), reference #19, references #21-22. The authors should change these references for other more appropriate references.

.- The last sentence of the second paragraph (“Specifically, a direct association has been observed between…”) should not be included here. This sentence is about the association between lipid levels and clinical endpoints, but in the previous sentence the authors are commenting about the association between genetic variation and metabolic dysfunction.

-. In the last paragraph the authors state that “…numerous SNP-associated dyslipidemia studies among HIV patients have come from non-African countries [18, 24, 25]”. However this is not true since the reference #24 is a study performed in a cohort of HIV patients from Zimbabwe.

2.- Materials and methods

.- In the section of sample size determination the authors assume an expected proportion of 0.1. Are there any previous study supporting this figure?.

.- In the data collection and biochemical analysis section the authors state that patients with triglycerides levels above 4.52 nmol/L were excluded from the study. The authors should explain to what extent this fact may impose a selection biass on the study population.

3.- Results

.- The proportion of females is much higher that the proportion of males. Is there any specific reason for this? Is this representative of the genre distribution among the HIV population in Ghana?. Moreover, the females proportion in the HIV-infected population is so high that the results obtained regarding the association between SNPs and dyslipidemia do really apply to only the female genre. The authors should discuss this fact and include it as one of the limitations of the study.

.- In table 1, immunovirological data of HIV population should be included (CD4 counts, time on HAART, pre-HAART HIV plasma viremia, etc..).

.- In the paragraph commenting the results given in table 2 the authors state that there was a significant difference (p=0.021) between cases and controls for rs662799 G/G genotype. However this p-value (as it appears on the table) is for the comparison of the frequency of allelic distribution and not of C/C genotype distribution. This is very confusing. Moreover, the authors must also compare the genotypes distribution and not only the allelic distribution and include this data in the table 2. The authors do not explain the reason to compare the allelic and genotypes distribution between HIV-infected and HIV-seronegative population. They do not comment anything about this comparison in the discussion section of the manuscript.

.-Table 3: data for these parameters in the control (HIV-seronegative) population should be given.

.- The way data are given on table 4 is somewhat confusing and not easy to quickly understand. A more understandable way to give the data would be to report the proportion of dyslipidemia in each group of patients according to the different SNPs genotypes. Moreover, when commenting the results given in this table, the authors state that “subjects with T/T genotype for LDLR polymorphism had a prevalence of dyslipidemia 3.46 times greater than non-dyslipidemic subjects.” Clearly this sentence makes no sense. Authors should pay more attention to the writing, since there are several errors either because a word is missing or because the word is not appropriate.

.- Data given in table 4 for HIV population should also be given for HIV-seronegative subjects.

.- Table 5: the authors report results from 3 different models. In each model the number of covariates differs, increasing from model 1 to model 3. Model 3 is the that includes the highest number of covariates and thus this is the only model that should be shown in table 5. In this table, the authors should include the definition of the terms: “Hypercholesterolemia”, “Hypertriglyceridemia”, “Low HDL-C”, and “High LDL-C”.

.- A table with the results of the multivariate logistic regression using dyslipidemia as the outcome variable should be included in the manuscript.

.-Supplementary data: a table comparing different characteristics between PI-base and NNRTI-based HAART regimens is shown as supplementary data. However there is no mention to this data in the text of the manuscript. The authors should briefly report in the text the findings of this supplementary data.

4.- Discussion

.- In the first sentence of the second paragraph the authors state that “the presence of….. among our study participants were very low..”Do the authors refer to the overall population of subjects included (HIV and HIV-seronegative together) of specifically to the HIV population. It is interesting that the prevalence of homozygous genotypes for the minor alleles (for ABCA1, DSCAML1 and APOA5 SNPs) is higher in HIV patients than in HIV-seronegative subjects, but the authors do not comment anything about this. They should discuss about this finding.

.- In the second sentence of the fifth paragraph the authors state that “This observation could be explained by the observation that almost half of the subjects carried the ABCA1 (rs2066714) G/G or A/G mutant……”. This is interpretation is incorrect since G/G and A/G carriers represented 36.6% of the population and this figure is far from “almost half”.

.- In the fourth sentence of the fifth paragraph the authors state that “In addition, Pollin et al..”In my opinion the sentence should start with “In contrast..” because the results of Pollin et al are different from the results reported in this manuscript.

.- In the first sentence of the sixth paragraph the word “homologous” has no sense at all. It must be changed to “homozygous”. Also in this same paragraph the sentence “The rs662799 is a SNP in the APOA5 gene” is not necessary since this has been already explained before throughout the text.

English grammar comment: I strongly recommend the manuscript to be revised by a native English-speaker.

6. PLOS authors have the option to publish the peer review history of their article (what does this mean?). If published, this will include your full peer review and any attached files.

Reviewer #1: No

Reviewer #2: No

---

## [Author Response · Author response to Decision Letter 0]

6 Nov 2019

Response to Editorial comments

Dear Editor-In-Chief

Manuscript tittle: Association of Single Nucleotide Polymorphisms with Dyslipidemia in Antiretroviral Exposed HIV Patients in a Ghanaian population

Article # PONE-D-19-22071

We sincerely thank the reviewers for the truly helpful comments to improve the quality of our manuscript. We have read through carefully, please find our detailed responses below each specific point

Reviewer #1: 

Though obviously suppressing HIV transmission and extending the life expectancy of HIV-infected patients, antiretroviral therapy (ART) also shows some side effects on human health. For example, long-term use of ART is associated with metabolic disturbances, such as dyslipidemia. In this study, the author studied the relationship of gene variants with dyslipidemia in HIV-infected patients receiving ART. Their data suggested that protease inhibitor treatment might be associated with higher levels of total cholesterol (TC), low-density lipoprotein cholesterol (LDL-C), and triglycerides [TG], compared with other ART treatment. In addition, four SNPs from four genes (ABCA1, LDLR, APOA5, and DSCAML1) are associated with dyslipidemia in HIV-infected patients receiving ART.

Major issues:

Comment: In the introduction, to easily make the reader understand the whole paper, the author had better explain why they selected these four SNPs to be studied.

Response: The authors are grateful for the comment. We have provided extra information the four SNPS selected for the study, role in lipid metabolism and its relationship with lipid abnormities in the introduction section

Comment: The author said “Significantly increased levels of total cholesterol (TC) low-density lipoprotein cholesterol (LDL-C) and triglycerides [TG] were observed in protease inhibitor-based (PI) treated case subjects compared to non-PI-based case subjects” in the abstract. However, the similar presentation is deficient in the result section. Please add the corresponding content in the result section.

Response: Thank you for the comment. We have provided the information on the comparison of general characteristics of patient based on treatment type in the results section revised manuscript, as table was submitted as supplement table (s1. Table 1) 

Comment: To make the reader better understand the relationship of different ART with dyslipidemia in HIV-infected patients, the author may as well describe the number of HIV-infected patients who receive PI or NNRTI.

Response: We have provided a table in the supplement information describing the number of HIV patients who received PI or NNRTI in the revised manuscript

Minor issues:

Comment: In the first paragraph of the result section, “All but HDL-cholesterol levels were significant levels in the control subjects compared to case subjects [ 1.44±0.33 vs. 0.99±0.53, p<0.0001].” should be “All but HDL-cholesterol levels were significantly lower levels in the control subjects compared to case subjects [ 1.44±0.33 vs. 0.99±0.53, p<0.0001]”

Response: We have revised this sentence in the new manuscript

Comment: In Table 5, the author should clarify the definition of PRR in the sentence “DSCAML1 (T/T) [PRR=11.46(3.65-43.56), p<0.0001)”.

Response: Results have been restructured therefore section have been revised for proper clarification

Reviewer #2: 

In the present study, Acheampong et al analyze the potential association of genetic variation in four different genes with the existence of dyslipidemia in HIV-infected patients on stable cART from Ghana.

Overall Major concern

The main concern of the study is that the design is very confusing. The authors state that the study is a case-control study. However there seems to be some confusion in the groups of selected individuals and their assignment to either cases or controls. In a case-control study, cases are those individuals presenting the disease (in this case dyslipidemia) and controls are those not presenting the disease. The objective of the study (as stated by the authors) is to “. investigate the distribution of SNPs in four candidate genes and resultant association with plasma lipid levels in Ghanaian HIV-infected patients on HAART”. For this purpose, cases should be HIV patients with dyislipidemia and controls HIV patients without dyslipidemia. However, the authors recruit a population of HIV-seronegative adults without dyislipidemia as controls (in the author’s own words: “…alongside age-matched control subjects with no history of HIV, dyslipidemia, hypertension, and diabetes.”). In my opinion this is not the ideal control population as I have explained above. The inclusion of a population of individuals seronegative for HIV (with and without dyslipidemia) would serve to answer the question: Are the SNPs associated with dyslipidemia in the HIV population the same as those in the non-HIV population? However, this is not the question that the objective addresses as the authors state (see above). Moreover, thereafter in the results section, the authors report that 18,3% of controls (ie. HIV-seronegative individuals) presented dyslipidemia. Thus, there is a contradiction between the results and the inclusion criteria for the control population (“age-matched control subjects with no history of HIV, dyslipidemia, hypertension, and diabetes”). Also, the only data given for the control population is the prevalence of dyslipidemia and the distribution of allelic and genotypes frequencies for the different SNPs, but no data are given about the association of SNPs and dyslipidemia in this control population. Lastly, there is no discussion at all about the findings in the HIV-seronegative (“control”) population. Thus, what is the reason to include this “control” population? n summary the authors must clarify this and explain the reason to include the “control” (HIV-seronegative individuals) population.

Response: We are grateful for the comment which has enabled us to have a critical look at our manuscript. Since knowledge on presence of the SNPS was not known in the general population for the selected genes, authors deemed it relevant to established. This is to strengthen our findings in the HIV-infected recruited of the study. The HIV seronegative individual recruited did not initially have any history of dyslipidaemia, diabetes or any chronic disease as stated in the manuscript, however after the measurement of lipid parameters and used of the NCEP-ATP III criteria to diagnose for dyslipidaemia, 18.3% were found to be dyslipidaemic. Based on the comments from the reviewer, we excluded the 18.3% HIV seronegative individual with dyslipidaemia from the data analysis. To further strengthen our findings, the HIV infected population were categorised into two groups, thus those with dyslipidaemia and those without dyslipidaemia. Therefore, all statistical analyses were made in these three groups. HIV seronegative individual without dyslipidaemia were used as a referent in statistical comparisons and association in the revised manuscript.

Specific concerns

1. Introduction:

Comment: Some references are not the most adequate, for example references #3-5, reference #1 (this reference is very old), reference #19, references #21-22. The authors should change these references for other more appropriate references.

Response: Thank you for the comments, we have updated our references as suggested by the reviewer.

Comment: The last sentence of the second paragraph (“Specifically, a direct association has been observed between…”) should not be included here. This sentence is about the association between lipid levels and clinical endpoints, but in the previous sentence the authors are commenting about the association between genetic variation and metabolic dysfunction.

Response: Thank you for the comment, we have excluded the sentence as suggested in the revised manuscript

Comment: In the last paragraph the authors state that “…numerous SNP-associated dyslipidaemia studies among HIV patients have come from non-African countries [18, 24, 25]”. However, this is not true since the reference #24 is a study performed in a cohort of HIV patients from Zimbabwe.

Response: Thank you for comment, sentence has been revised as studies on SNPS are few compared to the advanced countries

2. Materials and methods

Comment: In the section of sample size determination the authors assume an expected proportion of 0.1. Are there any previous study supporting this figure?

Response: Section has been revised; as it was based on previous report from a study conducted on the burden of dyslipidaemia among adults in Ghanaian a population and the lack of knowledge of the frequency of polymorphisms in the population. we speculate a value to give a strong statistical power. 

Comment: In the data collection and biochemical analysis section the authors state that patients with triglycerides levels above 4.52 nmol/L were excluded from the study. The authors should explain to what extent this fact may impose a selection bias on the study population.

Response: Section has been revised. We did not include ppatients with triglycerides above 4.52 mmol/L in the study since LDL-C was not directly measured and due to the deficit of the Friedewald’s equation, which overestimates LDL-C levels when triglycerides are high. This is to reduce any bias that might affect the relationship found between LDL-C and dyslipidaemia

3. Results

Comment: The proportion of females is much higher that the proportion of males. Is there any specific reason for this? Is this representative of the genre distribution among the HIV population in Ghana? Moreover, the female’s proportion in the HIV-infected population is so high that the results obtained regarding the association between SNPs and dyslipidemia do really apply to only the female genre. The authors should discuss this fact and include it as one of the limitations of the study.

Response: The authors are grateful for the comment. Previous reports have shown an uneven continuous rate of the incidence of HIV in the Ghanaian population with more than 60% of people living with HIV are females, nonetheless we have indicated in the revised manuscript as potential limitation.

Comment: In table 1, immunovirological data of HIV population should be included (CD4 counts, time on HAART, pre-HAART HIV plasma viremia, etc..).

Response: Section has been revised as CD4 counts has been included in the data analysis.

Comment: In the paragraph commenting the results given in table 2 the authors state that there was a significant difference (p=0.021) between cases and controls for rs662799 G/G genotype. However, this p-value (as it appears on the table) is for the comparison of the frequency of allelic distribution and not of C/C genotype distribution. This is very confusing. Moreover, the authors must also compare the genotypes distribution and not only the allelic distribution and include this data in the table 2. The authors do not explain the reason to compare the allelic and genotypes distribution between HIV-infected and HIV-seronegative population. They do not comment anything about this comparison in the discussion section of the manuscript.

Response: Thank you for the comment; we have restructured Table 2, where we have provided the Hardy-Weinberg equation (HWE)-p-value for the allelic frequency using HWE. Chi-square analysis has also been used to obtain p-value for the difference in genotype frequency in the different groups. Interpretation has been revised for clarity in the revised manuscript.

Comment: Table 3: data for these parameters in the control (HIV-seronegative) population should be given.

Response: The authors are grateful; data on the genotype in the HIV seronegative have been provided in the table in the revised manuscript

Comment: The way data are given on table 4 is somewhat confusing and not easy to quickly understand. A more understandable way to give the data would be to report the proportion of dyslipidemia in each group of patients according to the different SNPs genotypes. Moreover, when commenting the results given in this table, the authors state that “subjects with T/T genotype for LDLR polymorphism had a prevalence of dyslipidemia 3.46 times greater than non-dyslipidemic subjects.” Clearly this sentence makes no sense. Authors should pay more attention to the writing, since there are several errors either because a word is missing or because the word is not appropriate.

Response: Thank you for the comment, Table has been restructured in the revised manuscript and interpretation has been done for clarity

Comment: Data given in table 4 for HIV population should also be given for HIV-seronegative subjects.

Response: Thank you for the comment; data on the genotype in the HIV seronegative have been provided in Table 4 in the revised manuscript.

Comment: Table 5: the authors report results from 3 different models. In each model the number of covariates differs, increasing from model 1 to model 3. Model 3 is the that includes the highest number of covariates and thus this is the only model that should be shown in table 5. In this table, the authors should include the definition of the terms: “Hypercholesterolemia”, “Hypertriglyceridemia”, “Low HDL-C”, and “High LDL-C”.

Response: Thank you for the comment, as suggested by the reviewer, we have reported model 3 in the revised manuscript which other confounding factors were adjusted for. The NCEP-ATP III criteria used to define dyslipidaemia provides the various cut-off for high TC, low HDL-C, high LDL-C A and high TG were provided in this manuscript

Comment: A table with the results of the multivariate logistic regression using dyslipidaemia as the outcome variable should be included in the manuscript.

Response: We have provided a graph for the multivariate logistic regression using dyslipidaemia as a dependable variable generated with R program 

Comment: Supplementary data: a table comparing different characteristics between PI-base and NNRTI-based HAART regimens is shown as supplementary data. However, there is no mention to this data in the text of the manuscript. The authors should briefly report in the text the findings of this supplementary data.

Response: We have provided information on the supplement data on the table comparing different characteristics between PI-base and NNRTI-based HAART regimens

4. Discussion

Comment: In the first sentence of the second paragraph the authors state that “the presence of… among our study participants were very low ”Do the authors refer to the overall population of subjects included (HIV and HIV-seronegative together) of specifically to the HIV population. It is interesting that the prevalence of homozygous genotypes for the minor alleles (for ABCA1, DSCAML1 and APOA5 SNPs) is higher in HIV patients than in HIV-seronegative subjects, but the authors do not comment anything about this. They should discuss about this finding.

Response: Section has been revised in accordance with the reviewer comments.

Comment: In the second sentence of the fifth paragraph the authors state that “This observation could be explained by the observation that almost half of the subjects carried the ABCA1 (rs2066714) G/G or A/G mutant……”. This is interpretation is incorrect since G/G and A/G carriers represented 36.6% of the population and this figure is far from “almost half”.

Response: Thank you for the comment; section has been revised and proper interpretation has been done for clarity.

Comment: In the fourth sentence of the fifth paragraph the authors state that “In addition, Pollin et al.” In my opinion the sentence should start with “In contrast” because the results of Pollin et al. are different from the results reported in this manuscript.

Response: Response: Thank you for the comment; section has been revised

Comment: In the first sentence of the sixth paragraph the word “homologous” has no sense at all. It must be changed to “homozygous”. Also, in this same paragraph the sentence “The rs662799 is a SNP in the APOA5 gene” is not necessary since this has been already explained before throughout the text.

Response: Thank you for the comment; section has been revised

Comment: English grammar comment: I strongly recommend the manuscript to be revised by a native English-speaker.

Response: Thank you for the comment; section has been revised.

---

## [Decision Letter · Decision Letter 1]

29 Nov 2019

PONE-D-19-22071R1

Association of Single Nucleotide Polymorphisms with Dyslipidemia in Antiretroviral Exposed HIV Patients in a Ghanaian population

PLOS ONE

Dear Mr Acheampong,

Thank you for submitting your revised manuscript to PLOS ONE. After careful consideration, we feel that it has merit but does not fully meet PLOS ONE’s publication criteria as it currently stands. Therefore, we invite you to submit a second revised version of the manuscript that addresses the points raised during the review process. Please pay attention to the issues raised by the Reviewer #2, and properly answer her/his comments and revise the main text.

We would appreciate receiving your revised manuscript by Jan 13 2020 11:59PM. To enhance the reproducibility of your results, we recommend that if applicable you deposit your laboratory protocols in protocols.io, where a protocol can be assigned its own identifier (DOI) such that it can be cited independently in the future. For instructions see: http://journals.plos.org/plosone/s/submission-guidelines#loc-laboratory-protocols

We look forward to receiving your revised manuscript.

Kind regards,

Yong-Gang Yao

Academic Editor

PLOS ONE

Reviewers' comments:

Reviewer's Responses to Questions

**Comments to the Author**

1. If the authors have adequately addressed your comments raised in a previous round of review and you feel that this manuscript is now acceptable for publication, you may indicate that here to bypass the “Comments to the Author” section, enter your conflict of interest statement in the “Confidential to Editor” section, and submit your "Accept" recommendation.

Reviewer #1: All comments have been addressed

Reviewer #2: (No Response)

2. Is the manuscript technically sound, and do the data support the conclusions?

Reviewer #1: Yes

Reviewer #2: Partly

3. Has the statistical analysis been performed appropriately and rigorously? 

Reviewer #1: Yes

Reviewer #2: No

4. Have the authors made all data underlying the findings in their manuscript fully available?

Reviewer #1: No

Reviewer #2: Yes

5. Is the manuscript presented in an intelligible fashion and written in standard English?

Reviewer #1: Yes

Reviewer #2: No

6. Review Comments to the Author

Reviewer #1: The author has addressed all my questions. I don't have other major concerns. However, there is still a few minor issues. For example, in the introduction, "Low-density lipoprotein receptor (LDLR), positioned on chromosome 19p13.2 plays a significant role lipoprotein metabolism by mediating the uptake of cholesterol through

the binding and subsequent cellular uptake of apolipoprotein-E and B- constituting lipoproteins." should be "Low-density lipoprotein receptor (LDLR), positioned on chromosome 19p13.2 plays a significant role in lipoprotein metabolism by mediating the uptake of cholesterol through the binding and subsequent cellular uptake of apolipoprotein-E and B- constituting lipoproteins." So the author needs to proofread the whole manuscript carefully.

Reviewer #2: The revised version of the manuscript by Acheampong et al is still very far from being suitable for publication, for the next reasons:

1.- The authors did not satisfactorily answer the overall major concern regarding the design of the study and the rationale to include HIV-seronegative individuals. This is a study about the influence of several SNPs in the presence of dyslipidemia in HIV-patients on HAART. So, unless there is a very clear and strong reason as I explained in my first revision, HIV-seronegative individuals must be excluded from the study.

I strongly recommend the authors to exclude the 104 HIV-seronegative individuals from the study and reanalyze the data again only with the 289 HIV-patients.

2.- The design of the study is not clear at all and authors must clarify this. Is this a case-control study? (in my opinion it is). I already explained in my first revision who are the cases and who are the controls in a case/control study. If the authors do not consider that this is a case/control study, then the use of the terms “study groups”; “HIV+Dys+”; “HIV+Dys-“ is OK but all mention to “cases” and “controls” must be deleted from the manuscript.

I strongly recommend the help of a statistician.

3.- The way data are presented in tables 4 and 5 (logistic regression with diyslipidemia and with individual lipid abnormalities) and in figures 2 and 3, is incomprehensible. As the authors explain, the odds ratios shown in these tables and figures are referred to the “control” HIV-seronegative population. This is very difficult to understand. The clearest way to show evidence for association between the genotype of the different SNPs and the existence of dyslipidemia, is to calculate the odds ratios of having dyslipidemia in carriers of the minor homozygous genotype for each of the SNPs, taking the other individuals (carriers of the major homozygous and of the heterozygote genoypes) as reference (OR=1).

4.- The manuscript is still plagued with grammatical errors, sentences difficult to understand and words not appropriate, etc…

I strongly recommend that the whole manuscript is revised by a native English speaker

7. PLOS authors have the option to publish the peer review history of their article (what does this mean?). If published, this will include your full peer review and any attached files.

Reviewer #1: No

Reviewer #2: No

---

## [Author Response · Author response to Decision Letter 1]

23 Dec 2019

Response to Editorial comments

Dear Editor-In-Chief

Manuscript tittle: Association of Single Nucleotide Polymorphisms with Dyslipidemia in Antiretroviral Exposed HIV Patients in a Ghanaian population

Article # PONE-D-19-22071

We sincerely thank the reviewers for the truly helpful comments to improve the quality of our manuscript. We have read through carefully, please find our detailed responses below each specific point

Review Comments to the Author

Reviewer #1: 

Comment: The author has addressed all my questions. I don't have other major concerns. However, there is still a few minor issues. For example, in the introduction, "Low-density lipoprotein receptor (LDLR), positioned on chromosome 19p13.2 plays a significant role lipoprotein metabolism by mediating the uptake of cholesterol through the binding and subsequent cellular uptake of apolipoprotein-E and B- constituting lipoproteins." should be "Low-density lipoprotein receptor (LDLR), positioned on chromosome 19p13.2 plays a significant role in lipoprotein metabolism by mediating the uptake of cholesterol through the binding and subsequent cellular uptake of apolipoprotein-E and B- constituting lipoproteins." so the author needs to proofread the whole manuscript carefully.

Response: Thank you for the comment. Section has been revised and manuscript has reviewed by an English native speaker

Reviewer #2: 

The revised version of the manuscript by Acheampong et al is still very far from being suitable for publication, for the next reasons:

1. Comment: The authors did not satisfactorily answer the overall major concern regarding the design of the study and the rationale to include HIV-seronegative individuals. This is a study about the influence of several SNPs in the presence of dyslipidemia in HIV-patients on HAART. So, unless there is a very clear and strong reason as I explained in my first revision, HIV-seronegative individuals must be excluded from the study.

I strongly recommend the authors to exclude the 104 HIV-seronegative individuals from the study and reanalyse the data again only with the 289 HIV-patients.

Response: Thank you for the comment, the authors are grateful. We have excluded the 104 HIV sero-negative individuals from the study population and then reanalyse the dataset for the 289 HIV patients

2.Comment: The design of the study is not clear at all and authors must clarify this. Is this a case-control study? (in my opinion it is). I already explained in my first revision who are the cases and who are the controls in a case/control study. If the authors do not consider that this is a case/control study, then the use of the terms “study groups”; “HIV+Dys+”; “HIV+Dys-“ is OK but all mention to “cases” and “controls” must be deleted from the manuscript. I strongly recommend the help of a statistician.

Response: Thank you for the comment. We believed that base on the reviewer comment and it is case-control study, section has been revised. We have considered the study as case-control and therefore the terms “HIV+Dys+” and “HIV+Dys-” has been removed from the manuscript. A statistician was involved in analysing the data.

3. Comment: The way data are presented in tables 4 and 5 (logistic regression with diyslipidemia and with individual lipid abnormalities) and in figures 2 and 3, is incomprehensible. As the authors explain, the odds ratios shown in these tables and figures are referred to the “control” HIV-seronegative population. This is very difficult to understand. The clearest way to show evidence for association between the genotype of the different SNPs and the existence of dyslipidemia, is to calculate the odds ratios of having dyslipidemia in carriers of the minor homozygous genotype for each of the SNPs, taking the other individuals (carriers of the major homozygous and of the heterozygote genotypes) as reference (OR=1).

Response: Thank you for the comments. Tables 4 and 5 have been revised based on the reviewer’s comment. Figures 2 and 3 have removed. We have reanalysed the data by calculating the odd ratios taking into consideration the combination of the carriers of the major homozygous and the heterozygote genotypes as referent 

4.Coment: The manuscript is still plagued with grammatical errors, sentences difficult to understand and words not appropriate, etc…I strongly recommend that the whole manuscript is revised by a native English speaker

Response: Thank you for the comment. Our manuscript has been reviewed and revised by two native English speakers.

---

## [Editor Report · Decision Letter 2]

30 Dec 2019

Association of single nucleotide polymorphisms with dyslipidemia in antiretroviral exposed HIV patients in a Ghanaian population:  a case-control study

PONE-D-19-22071R2

Dear Dr. Acheampong,

We are pleased to inform you that your manuscript has been judged scientifically suitable for publication and will be formally accepted for publication once it complies with all outstanding technical requirements.

With kind regards,

Yong-Gang Yao

Academic Editor

PLOS ONE

---

## [Editor Report · Acceptance letter]

2 Jan 2020

PONE-D-19-22071R2 

Association of single nucleotide polymorphisms with dyslipidemia in antiretroviral exposed HIV patients in a Ghanaian population:  a case-control study 

Dear Dr. Acheampong:

I am pleased to inform you that your manuscript has been deemed suitable for publication in PLOS ONE. Congratulations! Your manuscript is now with our production department. 

With kind regards,

on behalf of

Dr. Yong-Gang Yao 

Academic Editor

PLOS ONE